# Maximising Lucerne (*Medicago sativa*) Pasture Intake of Dairy Cows: 2—The Effect of Post-Grazing Pasture Height and Mixed Ration Level

**DOI:** 10.3390/ani10050904

**Published:** 2020-05-22

**Authors:** Kieran A. D. Ison, Marcelo A. Benvenutti, David G. Mayer, Simon Quigley, David G. Barber

**Affiliations:** 1School of Agriculture and Food Sciences, The University of Queensland, Gatton Campus, Gatton QLD 4343, Australia; s.quigley@uq.edu.au; 2Queensland Department of Agriculture and Fisheries, Gatton Campus, Gatton QLD 4343, Australia; marcelo.benvenutti@daf.qld.gov.au (M.A.B.); david.mayer@daf.qld.gov.au (D.G.M.); david.barber@daf.qld.gov.au (D.G.B.)

**Keywords:** lucerne, grazing management, grazing dynamics, *Medicago sativa*

## Abstract

**Simple Summary:**

Pasture allocation has significant effects on grazing intensity, pasture utilisation and dry matter intake in grazing dairy herds. In sub-tropical Australian partial mixed ration (PMR), systems offering lucerne at an ideal pre-grazing pasture height and allocation, that ensures a proportion remains ungrazed, allows cows to selectively graze the top leafy stratum of the sward. When cows are only grazing the top leafy stratum, diet quality and intake is maximised in PMR systems irrespective of the amount of mixed ration offered.

**Abstract:**

The effects of lucerne (*Medicago sativa*) post-grazing residual pasture height on pasture utilisation (vertical and horizontal), pasture intake and animal production were investigated in a sub-tropical partial mixed ration dairy system. The study took place at the Gatton Research Dairy, Southeast Queensland (−27.552, 152.333), with a 26-day adaptation period followed by two 8-day measurement periods during August and September 2018. A quantity of 30 multiparous Holstein-Friesian dairy cows were offered two levels of mixed ration, 7 and 14 kg dry matter (DM)/cow/day for low and high levels respectively, and five levels of pasture allocation, to achieve decreasing residual pasture heights. Pasture allocations measured from 5 cm above ground level for the low mixed ration groups averaged 12.7, 15.9, 19.8, 35.3 and 49.2 kg DM/cow/day, and for the high mixed ration groups averaged 5.0, 8.3, 10.3, 18.6, and 25.2 kg DM/cow/day, respectively. As pasture allocation decreased, cows were forced to graze further down into the pasture sward, and therefore residual pasture height declined. Total intake (kg DM/cow/day) declined as residual pasture height (expressed as % of the initial height) declined, irrespective of mixed ration level, decreasing by 0.5 kg DM/cow/day for every 10% decrease in residual pasture height. Low total intakes were associated with high non-esterified fatty acid (NEFA) levels in plasma, indicating mobilisation of fat tissue to maintain milk production. In the high allocation treatments, an area of pasture remained ungrazed and cows were only grazing the top leafy stratum where pasture intake rate and intake were highest. Therefore, to maximise intake in sub-tropical partial mixed ration (PMR) systems, lucerne pasture should be allocated so that cows are always grazing the top leafy stratum. This can be achieved by ensuring the pasture around faecal patches remains ungrazed.

## 1. Introduction

Pasture allocation has significant effects on grazing intensity, post-grazing residue height and pasture utilisation and intake in beef [1,2] and dairy herds [3,4]. In the sub-tropical regions of Australia, an increasing number of dairy systems combine pasture and mixed rations, termed partial mixed ration (PMR) systems, to meet the nutritional requirements of the lactating herd [5]. Within these systems, pure lucerne pastures are used to provide a low-cost, high-quality source of forage. Developing grazing management strategies based on the ideal post-grazing pasture height that maximise intake and diet quality in lucerne pastures, could have significant impacts on the economics of sub-tropical partial mixed ration systems. Achieving high levels of pasture intake could limit the use of expensive mixed ration ingredients, or increase the milk response, and thus increase margin over feed cost.

Grazing management strategies that target high levels of utilisation have been shown to have negative impacts on pasture intake. Recently, Ison et al. [6] found that low allocations resulted in high levels of pasture utilization and lower intake in cows grazing lucerne in partial mixed ration systems. The study identified that cows selectively grazed the top leafy portion of lucerne across the whole area of the paddock, before regrazing the residual ‘stemmy’ portion of pasture remaining in previously grazed areas of their current paddock [6]. When grazing the top leafy stratum, both intake and diet quality were maximised, as cows removed larger bites from the top leafy stratum with a high leaf content relative to stem [6]. Once the top leafy stratum was depleted across the whole allocation, cows began to regraze areas of the paddock. In doing so, bite depth decreased and consequently pasture intake rate and intake decreased [6]. Therefore, allocating pastures based on the horizontal utilisation of the top leafy stratum, rather than vertical utilisation of the pasture, maximises intake. These results lead to the development of a new grazing management strategy, called proportion of ungrazed uncontaminated pasture (PUP) grazing, to maximise pasture intake. Pasture allocations based on this strategy allow the cows to graze the top leafy stratum, except around the faecal patches. This occurs because allocations are large enough to ensure cattle only graze across the area of a paddock once and can selectively remove the top leafy stratum to achieve daily intake requirements. When PUP approaches 0%, intake declines as cows shift from grazing the top leafy stratum to subsequent lower strata, where the pasture mass removed declines [6]. Achieving horizontal utilisations where the pasture around faecal patches remains ungrazed (“PUP strategy”), ensures cows are only grazing the top leafy stratum, and intake and diet quality are maximised. This can be achieved by allocating pasture based on the top leafy stratum on offer, excluding the proportion of the paddock contaminated with faecal patches. Therefore, the higher the amount of faecal patches, the greater the proportion of the total paddock area that should be left ungrazed to maximise pasture intake.

The first paper in this series “Maximising Lucerne pasture intake of dairy cows: 1—the effect of pre-grazing pasture height and partial mixed ration level”, investigated the effect of pre-grazing pasture height on lucerne pasture intake when pasture allocation was not limiting. The study focused on changes in pasture structure in the vegetative stage of growth (increasing levels of pre-grazing pasture height). Overall, the study identified that grazing lucerne at a pre-grazing pasture height below 40 cm had a negative impact on total intake, irrespective of mixed ration level. Following on from that work, it was identified that further research was required to determine how the post-grazing residual height impacts intake and production at different levels of mixed ration allocation. Furthermore, it is warranted to assess the effect on intake and milk production when using the PUP grazing strategy, compared to high vertical-utilisation grazing strategies, at the ideal pre-grazing pasture height identified in the first paper by [6].

The objective of this study was to test the following hypotheses: (1) pasture intake would decline as the residual pasture height declines; (2) mixed ration allocation would not affect total intake, irrespective of residual pasture height; and (3) milk production would increase as total intake increased.

## 2. Materials and Methods

The study took place at the Gatton Research Dairy, Southeast Queensland (−27.552, 152.333), with a 26-day adaptation period (6–31 August 2018) followed by two 8-day measurement periods (1–8 and 11–18 September 2018). Both periods were conducted in different areas of the same paddock. The current study was conducted in agreement with the guidelines of the Australian Code of Practice for the Care and Use of Animals for Scientific Purposes (National Health and Medical Research Council 2013) and was approved by the Department of Agriculture and Fisheries Animal Ethics Committee (number SA 2018/07/643).

### 2.1. Experimental Design

A quantity of 30 multiparous Holstein-Friesian (*Bos taurus*) lactating dairy cows were randomly allocated to 10 treatment groups based on a principal components summarisation of: days in milk (92 ± 57), milk yield (36.1 L/cow/day ± 10.5), milk fat concentration (3.92% ± 1.38%), milk protein concentration (3.26% ± 0.62%), somatic cell count (80,000 ± 35,000 cells/mL), live weight (610 kg ± 49), body condition score (4.7 ± 0.3; 1–8 scale) and parity (2.5 ± 0.9) (Table 1). Treatment groups consisted of the factorial combination of two levels of mixed ration allocations (7 and 14 kg dry matter (DM)/cow/day for low and high groups respectively), and five levels of pure lucerne pasture allocation to achieve the five desired post-grazing pasture heights. These groups were not formally replicated but spread approximately evenly across residual pasture heights for a response-surface analysis.

The lucerne pasture sward had been established for two years and managed under typical commercial practices for the sub-tropical region [7]. The pasture was fertilised and irrigated to meet the pasture’s nutrient requirements and maintain soil moisture to maximise growth [7]. Bloat oil (271 g/L alcohols, C12–15 ethoxylated) was added daily to water troughs in each paddock at a rate of 42 mL/cow to reduce the risk of bloating. Incremental residual pasture heights were achieved by increasing the area allocated to each group. Area grazed targets were included to identify any contrast in intake between levels below and above the PUP threshold (~5% ungrazed pasture). Pasture allocation groups were assigned to make an easier comparison between mixed ration levels, as actual area allocations were different for all groups (Table 1). The accompanying paper [8] identified that pasture intake increased as pre-grazing height increased to 39 cm. Therefore, the targeted pre-grazing pasture height in this experiment was ~40 cm to ensure pasture structure (height) did not limit intake. Pasture allocations were determined using targeted residual heights, which were expected to result in different proportions of the pasture area left ungrazed, detailed in Section 2.2. The lower the pasture allocation, the lower the residual pasture height and the lower the pasture area left ungrazed. The two high pasture allocation treatments (groups 4 and 5; Table 1) were expected to result in similar residual pasture heights, but with a greater proportion of ungrazed pasture for group 5.

Cows were offered a mixed ration on a feed pad following the a.m. milking at 0800 h until 1000 h and all groups were moved to a pasture strip. At 1530 h all cows were milked, and at 1700 h cows in high mixed ration groups were returned to the feedpad to consume any remaining mixed ration, whilst the low mixed ration groups returned to their respective pasture strips. At 1900 h, high mixed ration cows were returned to their pasture strips and all groups grazed overnight until the following morning milking at 0630 h. Each pasture strip was divided into two sections to minimize trampling and spoiling of the pasture strip. Cows were offered one third of the allocation during the a.m. grazing period, and when returned to the pasture strip after the p.m. milking, the dividing fence was removed and cows had access to the entire allocation.

Both mixed rations were formulated using the Nittany Cow ration formulation program (NittanyCow, ‘Dairy Ration Evaluator’ Software, Mifflinburg, PA, USA). Diets were balanced to meet the metabolic requirements of cows producing an average of 25 L/cow/day, when combined with the expected nutrient intake of either 7 or 14 kg DM of the lucerne pasture for high and low mixed ration levels, respectively (Table 2). Note that actual pasture intake could not have been predicted before the experiment as it was not known; therefore, formulating a diet for each pasture allocation treatment was not possible. Daily mixed ration refusals were removed and weighed to calculate individual daily mixed ration intake as an average for each group of three cows. Pasture intake was calculated as an average for each group, detailed in Section 2.2.

### 2.2. Pasture Allocation, Dry Matter Intake and Diet Quality

Pasture allocation, structure, intake and quality were determined by calculating the top-down vertical distribution of DM and chemical composition of the lucerne pasture. This was determined using a random, stratified double-sampling method, adapted from Benvenutti et al. [2] and used by Ison et al. [6], to estimate and explain nutrient intake from pasture. A quantity of 22 pasture samples (ranging in height from 25 to 65 cm) were cut 5 cm above ground level, then further cut into four equal vertical strata. Pasture samples were taken at the beginning, middle and end of each experiment period. Within each stratum, DM (g) was determined by drying samples in ovens at 60 °C. Sub-samples from within each stratum were used to estimate mincing energy (Section 2.3), and analysed at Dairy One Forage Lab (Ithaca, NY, USA) to estimate crude protein (CP) % DM [9], neutral detergent fibre (NDF) and acid detergent fibre (ADF) % DM [10], and lignin, starch and sugar % DM [11]. Concentrations of metabolisable energy (ME) were calculated using the following formula [12]:ME (MJ/kg DM) = (((1.01 × (0.04409 × TDN)) − 0.45) × 4.184(1)
where TDN is total digestible nutrient (%). TDN was estimated using the following formula [12]:TDN = 5.31 + 0.412 CP% + 1.444 × Ether Extract% + 0.937 × Nitrogen Free Extract%(2)
where Ether Extract is calculated according to Randall/Soxtec/Submersion Method [13].

Each combination of targeted area grazed and residual pasture height was assigned a pasture allocation group: 1, 2, 3, 4 and 5, outlined in Table 1. Pasture was allocated to each of the higher-allocation treatment groups targeting higher pasture residue heights with an associated area of grazed pasture (horizontal utilisation) less than 100% of the paddock area (Table 1). This was achieved using the estimated intake from grazing the top leafy stratum at a pre-grazing pasture height of 40 cm. Ison et al. [6] identified that the grazing depth of the top leafy stratum was consistent for all groups, at an average of 15 cm. Therefore, the targeted residual height in allocations where cows were only grazing the top leafy stratum would be 15 cm below the pre-grazing pasture height. Pasture allocation groups 4 and 5 were offered large areas to ensure they were only grazing the top leafy stratum (~15 cm), and therefore had a targeted post-grazing residual height of 25 cm (Table 1). Each of these allocations were expected to result in the PUP grazing strategy as describe by Ison et al. [6]. The remaining lower-allocation groups (1, 2 and 3) were all targeted to achieve lower pasture residual heights and a result in horizontal utilisations of 100% of the grazable area. The lowest pasture allocation group (1) for each mixed ration level was determined by expecting cows to consume all pasture down to 5 cm above ground level to replicate current industry targets (Table 1). Two intermediate post-grazing residual height targets (groups 2 and 3) were calculated as even differences between the 5 and 25 cm residual height targets (Table 1). During the initial adaptation period, pasture allocations were restricted for each pasture allocation level to force cows to graze down to the targeted residual pasture height. During the final adaptation and measurement periods, pasture allocations were kept constant.

### 2.3. Mincing Energy

Lucerne stratum samples were minced using a method described by Chaves et al. [14] to achieve a particle size distribution similar to forages chewed by ruminants [15,16]. Samples were frozen at −16 °C and minced using a meat mincer (Reber 9500 N, Via Sanguine, 46,030 Correggioverde di Dosolo, Italy). The energy required for mincing was recorded using the method described by Sun et al. [17]. A kilowatt hour (kWh) meter (Cabac Power-Mate 10 AHD, Computer Control Instrumentations, Adelaide, South Australia, Australia) measured energy consumption and was logged by data logger software (Power-Mate Communicator 2.0.1, Computer Control Instrumentations, Adelaide, South Australia, Australia). The baseline power consumption was recorded and net energy in kJ for mincing 250 g of fresh forage was determined. DM for each sample was estimated from a subsample of the fresh material and from the dried minced material. DM estimations from the minced material were used to express net energy consumption in terms of kJ/g DM. The variation in mincing energy down the vertical plane of lucerne pasture was determined using calibration equations described in Section 2.2.

### 2.4. Defoliation Intensity and Pasture Intake

Defoliation intensity was defined by calculating the vertical and horizontal utilisation of the pasture. Height measurements were taken at fixed assessment points along transects within each grazing strip to allow more accurate measurement of the vertical difference in pasture height pre- and post-grazing [1,2,6]. Transects were taken every 2 m across the grazing strip, and measurements of pasture height and condition were recorded every 1 m along each transect, with approximately 100 measurement points per strip per day. The condition of the pasture was assessed at each point visually, a decline in pasture height indicated ‘grazed’, and no change, ‘ungrazed’. All assessment points were inspected for faeces and were recorded as ‘contaminated’ if present. Areas were identified and recorded as ‘trampled’ when pasture had been flattened onto the soil surface by cows walking or lying down. Trampled and contaminated points were later used to determine the ‘ungrazable’ area within each strip, as cows cannot bite into trampled pasture and avoid grazing areas contaminated with faeces [2]. The ungrazed area was subtracted from the total area allocated to determine the maximum ‘grazable’ area within each strip. Using the vertical distribution of DM and chemical composition of the pasture outlined in Section 2.2, pasture intake and quality was calculated as described by Ison et al. [6]. The combined horizontal and vertical utilisation of the grazed pasture was used to calculate the average intake for each treatment group every day. The average residual pasture height of the whole paddock was calculated as the average proportional pasture height, where:Residual pasture height (% of initial) = Residual pasture height (cm)/Pre-grazing pasture height (cm) × 100(3)

All grazed and ungrazed measurement points from the grazable areas within each pasture strip are expressed as residual pasture height (% of initial). This allows easier comparison of results between groups, as pre-grazing pasture height (cm) varied on average from 39 to 41 cm throughout the measurement period.

### 2.5. Ingestive Behaviour

During the measurement period all cows were equipped with RumiWatch halters (version 6.0) and pedometers (Itin + Hoch GmbH, Fütterungstechnik, Liestal, Switzerland; [18,19]). Data was converted with the RumiWatch Converter (C31; Itin + Hoch GmbH, Converter 0.7.3.31, Liestal, Switzerland). During the adaptation period, cows were fitted with lead halters and ankle tags for 5 days to become accustomed to the halter and pedometer. Cows were also fitted with the RumiWatch halters and pedometers for 3 days prior to the measurement period to allow each unit to calibrate to each animal and to check data was logging correctly [19]. The halters and pedometers were removed between measurement periods to avoid skin alterations. Eating time while grazing or consuming the mixed ration was recorded every day during each measurement period [18,20]. Location and time of all treatment groups was recorded during the measurement periods to determine when cows were on the feed pad (with access to the mixed ration and water only), milking (access to water only), on the pasture strip (access to pasture and water only) and in transit between locations (no access to feed or water). Eating time data was checked against manual recordings and combined with pasture and mixed ration intakes to estimate mixed ration eating time (h/cow/day), average intake rate of mixed ration (kg DM/h), pasture eating time (h/cow/day), average intake rate of pasture (kg DM/h), and combined total eating time (h/cow/day) and total average intake rate (kg DM/h).

### 2.6. Blood Collection and Analysis

Blood samples (20 mL) were collected from the jugular vein of individual cows on the final day of each measurement period (8 and 18 September 2018) using an 18 gauge needle and 20 mL syringe. Plasma was separated using a refrigerated centrifuge at 1200 rpm for 10 min (MSE Harrier 18/80, Heatherfield, East Sussex, UK), frozen and analysed by the University of Queensland Veterinary Pathology Laboratory for blood non-esterified fatty acids (NEFA) and β-hydroxybutyrate (BHBA).

### 2.7. Milk Production

Milk yield was measured at each milking period for individual cows by using automatic flow meters (Westfalia, ‘Dairy Plan’ Software, Düsseldorf, Germany), and samples were analysed for fat and protein (Siliker Australia, Eagle Farm, Brisbane Queensland, Australia). Energy-corrected milk, standardised to 4.0% fat and 3.3% protein, was calculated using the following formula [21]:Energy corrected milk (kg/cow/day) = milk yield (kg) × (376 × fat% + 209 × protein% + 948)/3138(4)

### 2.8. Statistical Analysis

All analyses were conducted using the GenStat^®^ (18th Edition, VSN International Ltd. Hemel Hempstead, Hertfordshire, England) software package. Multiple linear regressions were used to establish the vertical distribution calibration equations for pasture mass and quality. The backward (step-down) selection method was used, and explanatory variables were removed if *p* > 0.05. General linear models were used to assess the relationships between the explanatory (X) variables (mixed ration level and residual pasture height as a linear contrast), and the following response (Y) variables: pasture offered, pasture intake, mixed ration offered, mixed ration intake, total DM offered, total DM intake, quality parameters (ME, CP, NDF, ADF, Lignin, Starch, Sugars) for pasture, mixed ration and total diet as both kg DM/cow/day and % of the whole diet, milk, energy-corrected milk and milk fat and protein yield. The independent experimental units for these analyses were the groups (each with three cows as the subsamples). Preliminary analyses showed that there were no significant interactions between the design factors (mixed ration and residual pasture height) and measurement periods; therefore, cross-period results are presented.

## 3. Results

### 3.1. Defoliation Intensity

High mixed ration groups’ pasture allocations (kg DM/cow/day) increased linearly, from 5.0 to 25.2 for pasture allocation levels 1 to 5, and low mixed ration groups increased from 12.67 to 49.2 for pasture allocation levels 1 to 5 respectively (Table 3). Pasture allocation groups 1, 2 and 3 for both mixed ration levels grazed over 97% of the allocated area, and both 4 and 5 pasture allocation groups had at least 4.7% of the area ungrazed (Table 3). Pre-grazing pasture height was not significantly different between groups and averaged 40.3 cm above ground level. The average residual pasture height of the grazed areas declined as pasture allocation declined, however the minimum grazed residual pasture height was 14.7 and 13.8 cm for high and low mixed ration levels, respectively. The average residual pasture height (% of initial) declined linearly as pasture allocation declined (Table 3). Residual pasture height (% of initial) provides an indication of the grazing intensity for all treatment groups, as it combines both vertical utilisation and horizontal utilisation of the pasture sward. All data were analysed against residual pasture height (% of initial) as regressions, to show the relationships as grazing intensity increases.

### 3.2. Dry matter Intake and Diet Quality

Pasture intakes (Figure 1) had significant positive linear relationships with residual pasture heights and were significantly different between mixed ration levels (*p* < 0.05). Total intake had a significant positive linear relationship with residual pasture height (*p* < 0.05) and tended to be higher for high mixed ration levels compared to low mixed ration levels, averaging 21.1 and 19.4 kg DM/cow/day, respectively (*p* = 0.08; Figure 2).

The quality (CP % DM; ME, MJ/kg DM; NDF % DM and ADF % DM) of the consumed pasture is outlined in Table 4. The CP and ME of the consumed pasture had significant positive relationships with residual pasture height (*p* < 0.001) and were not different between mixed ration levels (Table 4). NDF and ADF had significant negative relationships with residual pasture height (*p* < 0.001) and were not different between mixed ration levels (Table 4). Lignin, starch and sugars of the consumed pasture did not vary significantly down the vertical plane of the pasture sward, averaging 8.5, 0.6 and 5.9 % DM respectively for all groups.

Pasture CP (kg DM/cow/day) and ME (MJ/cow/day) intake had significant positive relationships with residual pasture height (*p* < 0.001) and had a significant interaction between mixed ration level and residual pasture height (*p* < 0.001) (Table 5). NDF and ADF intakes (kg DM/cow/day) did not have a significant relationship with residual pasture height, but were significantly higher for the low mixed ration level (*p* < 0.001) (Table 5). Lignin, starch and sugar intakes had significant positive linear relationships with residual pasture height, and were significantly higher for the low mixed ration level (*p* < 0.001) (Table 5).

Mixed ration intake did not have a significant relationship with residual pasture height (*p* > 0.05) and was significantly different between mixed ration levels (*p* < 0.05), averaging 15.3 and 7.4 kg DM/cow/day, for high and low levels, respectively (Table 6). Mixed ration offered (kg DM/cow/day), intake (kg DM/cow/day) and nutrient (ME, MJ/cow/day; CP, NDF, ADF, lignin, starch and sugars in kg/cow/day) intakes did not have significant relationships with residual pasture height, and were expectantly higher for the high mixed ration level (*p* < 0.001) (Table 6).

Total intake (kg DM/cow/day), CP (kg DM/cow/day) and ME (MJ/cow/day) intake increased with increasing residual pasture height, and were significantly different between mixed ration levels (*p* < 0.01) (Table 7). There was no difference in total NDF or ADF intake (kg DM/cow/day) between all treatment groups. Total lignin and sugar intake (kg DM/cow/day) had a positive significant relationship with residual pasture height. Lignin was significantly higher for low mixed ration level (*p* < 0.01) (Table 7) and there was no difference between mixed ration levels for sugar intake. Starch intake (kg DM/cow/day) did not have a significant relationship with residual pasture height, but was significantly lower for low mixed ration level (*p* < 0.001) (Table 7).

The proportion of total diet comprised of pasture increased as residual pasture height increased, and was higher for the low mixed ration level (*p* < 0.01)(Table 8). CP % of the total diet increased as residual pasture height increased and was higher for the low mixed ration level, and there was a significant interaction between residual pasture height and mixed ration level (*p* < 0.01). ME (MJ/kg DM) increased as residual pasture height increased (*p* < 0.01) (Table 8), but was not different between mixed ration levels. NDF and ADF (%) of the total diet decreased as residual pasture height increased, and was significantly higher for low mixed ration level (*p* < 0.001) (Table 8). Lignin (%) of the total diet increased as residual pasture height increased, and was significantly higher for the low mixed ration level (*p* < 0.05) (Table 8). Starch (%) of the total diet decreased significantly as residual pasture height increased (*p* < 0.05) and was significantly higher for high mixed ration level (*p* < 0.001) (Table 8). Sugar (%) of the total diet did not have a significant relationship with residual pasture height, but was but was significantly higher for the low mixed ration level (*p* < 0.05) (Table 8).

### 3.3. Mincing Energy

Mincing energy (kJ/kg DM) of the consumed pasture had a significant negative linear relationship with residual pasture height (*p* < 0.05), and was not different between mixed ration levels (Figure 3).

### 3.4. Ingestive Behaviour

Grazing and total eating time did not vary significantly between mixed ration levels (*p* > 0.05), averaging 6.6 and 7.7 h/cow/day. Average daily pasture intake rate had a positive linear relationship with residual pasture height, and tended to be lower for the high mixed ration level, averaging 1.0 kg DM intake/cow.h compared to 1.7 kg DM intake/cow.h for the low mixed ration level (*p* = 0.13; Figure 4). Average daily mixed ration intake rate tended to increase as residual pasture height declined (*p* = 0.089), and tended to be higher for the high mixed ration level, averaging 14.2 kg DM intake/cow.h compared to 11.2 kg DM intake/cow.h for the low mixed ration level (*p* = 0.13, data not shown). Average intake rate for the whole diet tended to increase as residual pasture height increased (*p* = 0.14), and tended to be higher for the high mixed ration level: 2.9 kg DM intake/cow.h, compared to 2.4 kg DMI/cow.h for the low mixed ration level (*p* = 0.06; data not shown).

### 3.5. Plasma Metabolites

NEFA (mEq/L) had a significant negative linear relationship with residual pasture height (*p* < 0.05), and was not different between mixed ration levels (Figure 5). BHBA (mmol/L) had a significant positive linear relationship with residual pasture height (*p* < 0.05) and was significantly higher for the low mixed ration level (*p* < 0.05) (Figure 6).

### 3.6. Milk Production

Milk (L/cow/day) and energy-corrected milk (kg/cow/day) yield did not have a significant relationship with residual pasture height and was not significantly different between mixed ration levels (Figure 7 and Figure 8 respectively). Milk fat (%) did not have a significant relationship with residual pasture height, and tended to be lower for high mixed ration level compared to low mixed ration level, averaging 2.5 and 3.5 (%) respectively (*p* = 0.08; Figure 9). Milk protein had significant negative linear relationship with residual pasture height, and was not significantly different between mixed ration levels (Figure 10).

### 3.7. Tables Summarising Regression Statistics

The summary statics of the Figure 1, Figure 2, Figure 3, Figure 4, Figure 5, Figure 6, Figure 7, Figure 8, Figure 9 and Figure 10 in Section 3.1, Section 3.2, Section 3.3, Section 3.4, Section 3.5 and Section 3.6 are shown in Table 9.

## 4. Discussion

This study examined the effects of reducing pasture allocation to decrease residual pasture height, at two levels of mixed ration, on defoliation intensity, ingestive behaviour, post-grazing proportion of ungrazed pasture, total intake, and milk production and composition, for lactating dairy cows. Since the PUP strategy has been shown to drive pasture intake in grazing systems [1,2,6], the results are discussed below in the context of the PUP grazing strategy. PUP has been defined as a grazing strategy to maximise pasture intake in PMR dairy systems [6]. This is achieved by allocating pasture areas large enough to ensure cows selectively consume the top leafy stratum. Cows will remove the top leafy stratum across the whole area of a paddock before regrazing an area, and intake declines. This transition can be observed when cows begin to graze around faecal patches.

Pre-grazing height in this trial averaged 40.3 cm above ground level. The companion study by Ison et al. [8] indicated that grazing lucerne below 39 cm of pre-grazing height significantly reduced pasture intake. Furthermore, there was no difference in pre-grazing height between treatment groups, and therefore differences in defoliation dynamics and grazing intensity were caused by differences in pasture allocation. In the current study, the pasture allocation of the low allocation treatments was reduced in the adaptation period to encourage cows to graze down into the pasture sward, to achieve the targeted post-grazing pasture height for each group. The minimum post-grazing pasture height achieved by any treatment group throughout the measurement periods was 13.7 cm. This shows that, despite the low pasture allocation, cows refused to graze down into the sward, and explains the lower pasture intakes at low residual pasture heights. When allocations were high, as in pasture allocation groups 4 and 5, an area greater than 4% of the allocation always remained ungrazed. These cows were therefore always grazing the top leafy stratum [6]. When grazing the top leafy stratum, the pasture mass removed is greatest [1,2,6], and consequently these groups achieved high levels of pasture intake. When intakes of pasture and mixed ration were combined, total intake increased as pasture intake increased for both mixed ration levels. This indicates that using a strategy to allocate pastures to ensure some pasture remains ungrazed—the PUP grazing strategy—will maximise intake in PMR systems irrespective of the proportion of mixed ration in the diet. This reduction of pasture intake with decreasing post-grazing pasture residue height has been well documented by previous studies [1,2,6,23]. The novelty of our results is that the PUP can be used as a simple grazing management strategy, to allow the cows to graze the top leafy stratum and achieve high levels of pasture intake. Our results also show that PUP works irrespective of the level of mixed ration.

Not only did pasture allocation affect intake, but also diet quality declined significantly for groups that grazed down into the lower strata of the swards, which contain a greater proportion of stem. The mincing energy (kJ/kg DMI) of the consumed pasture increased linearly as residual pasture height decreased. This may explain why cows refused to graze down into the pasture sward and intake declined. Mincing energy provides an indication of the toughness of plant material, and shows a strong correlation with increased stem density and grazing resistance [22]. As cows graze further down into the lucerne sward, mincing energy, stem density and grazing resistance increase and cause a physical barrier to prehension and ingestion of plant material [22,23,24,25]. This may also explain why pasture intake rate declined as residual pasture height declined. Cows remove less material per bite when the top leafy stratum has been removed [6], and consequently intake declines. Cows could increase grazing time to maintain intake when intake rate declines, however, there was no difference in grazing time for all treatment groups. This further illustrates that cows avoided grazing down into the pasture sward when mincing energy and grazing resistance increased, despite not meeting targeted intakes. Diet quality was also unlikely to limit intake, as there was no difference in total diet NDF and ADF intake between all treatment groups. The average NDF intake was 5.54 kg DM/cow/day, only 0.9% of total body weight, and therefore it is unlikely that cows reached full rumen distention to limit intake [26]. ADF intake tended to increase, and lignin intake increased significantly as residual pasture height declined. This may explain some of the decline in intake, as increasing ADF and lignin was correlated with increasing mincing energy. As mincing energy decreased for the high post-grazing residues, forage particles were broken down more quickly in the rumen and passed through into the remaining digestive tract more quickly [12]. As passage rate increases, intake increases too, due to decreasing rumen distension. ME and CP content of the total diet increased as residual pasture height decreased. This was caused by the increase in both pasture quality and intake. Various studies have shown that lucerne pasture quality is higher in the leaves than the stems, and that leaves are more easily ingested [6,27]. This explains why quality and intake is higher when cows are grazing the top leafy stratum of lucerne pasture. Overall, total intake was maximised when pasture intake was maximised. Pasture intake increased due to high pasture intake rates, achieved when cows were grazing the top leafy stratum and grazing resistance was likely lowest, as indicated by mincing energy. Pasture intake was measured using a pasture-based method which has an error associated with it. It is unlikely that this error was a possible reason for the lack of difference in milk production between pasture treatments (i.e., due to intake being overestimated). If the methods overestimates pasture intake, it is likely that that the overestimation would apply for all treatments.

Milk fat tended to be higher for low mixed ration groups, averaging 3.5 %, versus 2.5 % for cows in high mixed ration groups. The low milk fat % in high mixed ration groups is likely explained by the higher proportion of starch and sugars in the diet. Starch increased from 24.4% to 30.1% of the total diet of the high mixed ration groups, as pasture intake declined from 7.9 to 3.6 kg DM/cow/day. National Research Council [12] recommends that starch comprises 25% of the diet, to provide a balance between energy density and rumen health, and to maximise yield without compromising milk quality through milk fat suppression. Only cows that achieved their targeted pasture intakes of 7 kg DM/cow/day had starch levels below the 25% threshold. This issue in diet composition could not have been avoided, as all diets were balanced based off the combined targeted intakes of mixed ration and pasture. This shows the importance of being able to accurately measure the intake and quality of pasture within PMR systems, to ensure a mixed ration can be formulated to meet the required nutrient supply for lactating cows without compromising production. Milk protein also had a small significant relationship with residual pasture height, which was likely caused by the higher intake of crude protein achieved by cows with the high pasture allocations. Although the PUP strategy did not have a significant effect on milk yield, it did influence milk composition by ensuring targeted intakes were achieved and diet quality was maximised. We hypothesised that milk production would increase as intake increased. However, despite the higher intakes at higher residual pasture heights, there were no significant effects on milk yield (L/cow/day) or energy-corrected milk yield (kg/cow/day).

Although production was not affected by pasture allocation and consequently intake, the blood analyses indicated that body fat reserves had been mobilised and used to support milk production. NEFA (mEq/L) increased significantly as residual pasture height declined for both mixed ration levels, indicating that fat tissue was mobilised [28]. When the mobilised body fat cannot be converted into energy through the usual pathways, ketone bodies such as BHBA are produced. Since BHBA did not increase in this study when residual pasture height decreased, it is likely that the mobilised fat was used as a source of energy to produce milk. This explains why milk yield did not decline with decreasing pasture allocation and intake. Fat mobilisation is likely to have a long-term negative impact on body condition and health, as well as reproductive performance, of the herd [28].

## 5. Conclusions

This study showed that utilising the PUP grazing strategy maximises total intake in a sub-tropical PMR dairy system. Although milk yield differences were not observed, blood analysis indicated that body reserves had been mobilised to maintain production when energy requirements for lactation had not been achieved. The negative impacts of body reserve mobilisation are likely to exacerbate over long periods, with potentially significant herd-health and economic impacts.

The ideal post-grazing target for lucerne pastures when grazed at an ideal pre-grazing height (>39 cm above ground level, identified in the companion paper [8]), should ensure the pasture around faecal patches remains ungrazed. Therefore, the higher the amount of faecal patches, the greater the proportion of the total paddock area that should be left ungrazed to maximise pasture intake. This ensure cows are selectively removing the top leafy stratum only, maximising intake and diet quality irrespective of the amount of mixed ration offered.

Further investigation is required to understand the long-term impacts on pasture persistence and quality. Secondary herds or mechanical methods could be utilised to reduce the residual pasture back to ideal agronomic levels. Economic analysis is required to understand the full potential of this strategy within commercial systems.

Finally, this study has shown how pasture grazing strategies can be manipulated, to increase pasture and consequently total intake with different levels of mixed ration intake. However, further investigation is warranted to assess how mixed ration quality can affect total intake when the PUP grazing strategy is implemented on sub-tropical PMR dairy systems.

## Figures and Tables

**Figure 1 animals-10-00904-f001:**
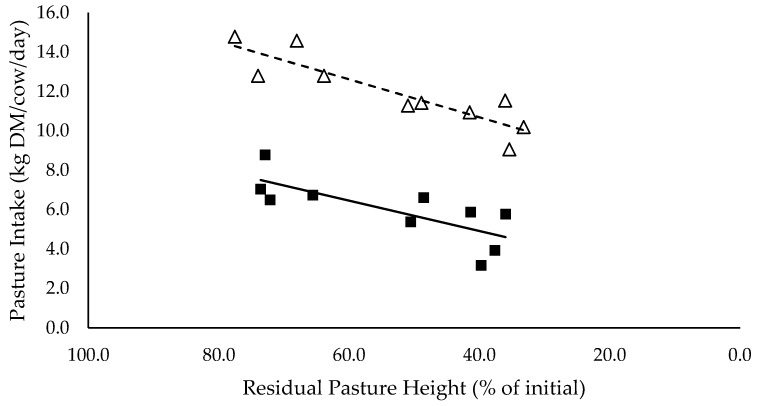
The relationship between pasture intake (kg DM/cow/day) and residual pasture height (% of initial) for high (■) and low (Δ) mixed ration levels. Lines were fitted for high (solid) and low (dashed) mixed ration levels.

**Figure 2 animals-10-00904-f002:**
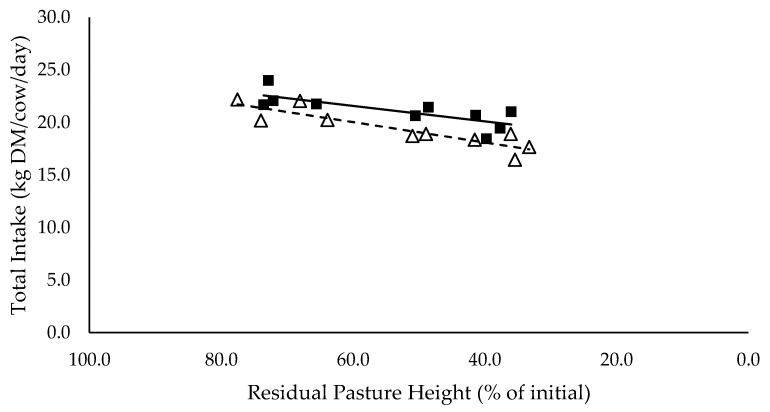
The relationship between total intake (kg DM/cow/day) and residual pasture height (% of initial) for high (■) and low (Δ) mixed ration levels. Lines were fitted for high (solid) and low (dashed) mixed ration levels.

**Figure 3 animals-10-00904-f003:**
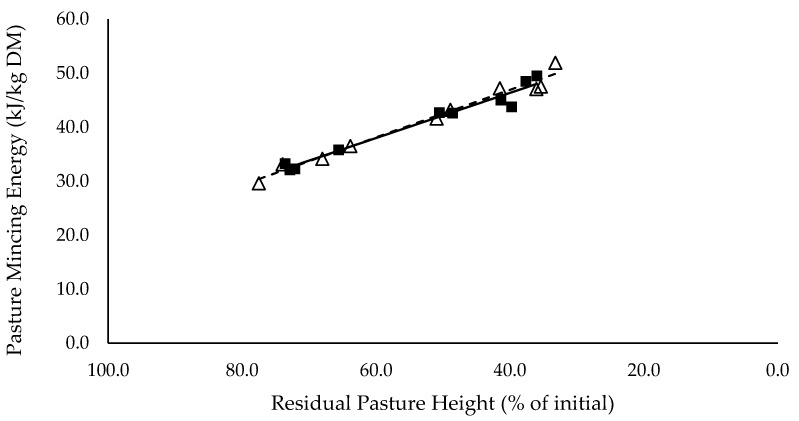
The relationship between average pasture mincing energy (kJ/kg DM) and residual pasture height (% of initial) for high (■) and low (Δ) mixed ration levels. Lines were fitted for high (solid) and low (dashed) mixed ration levels.

**Figure 4 animals-10-00904-f004:**
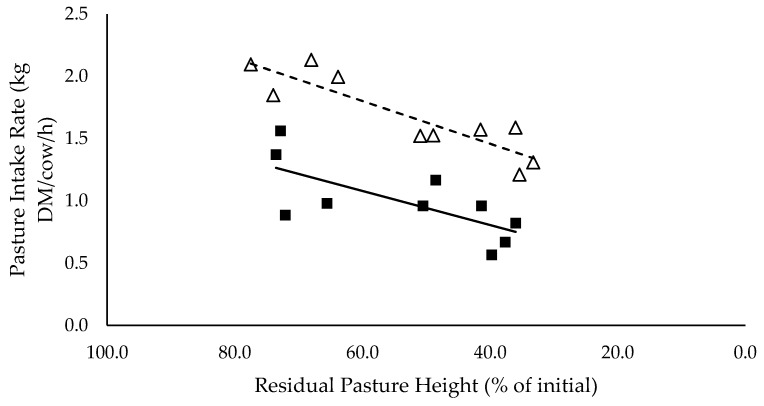
The relationship between pasture intake rate (kg DM/cow/h) and residual pasture height (% of initial) for high (■) and low (Δ) mixed ration levels. Lines were fitted for high (solid) and low (dashed) mixed ration levels.

**Figure 5 animals-10-00904-f005:**
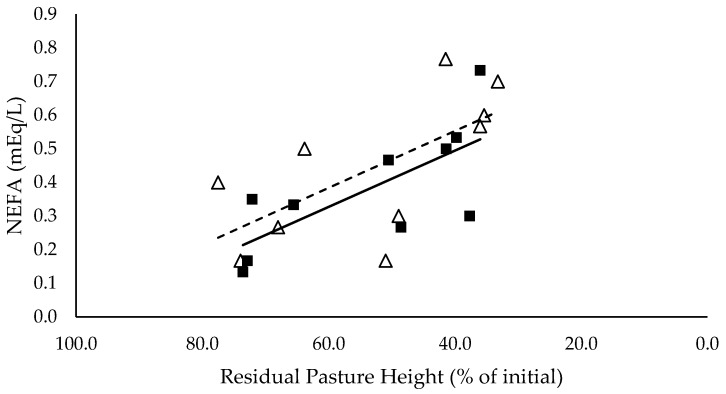
The relationship between the serum non-esterified fatty acid (NEFA) (mEq/L) and residual pasture height (% of initial) for high (■) and low (Δ) mixed ration levels. Lines were fitted for high (solid) and low (dashed) mixed ration levels.

**Figure 6 animals-10-00904-f006:**
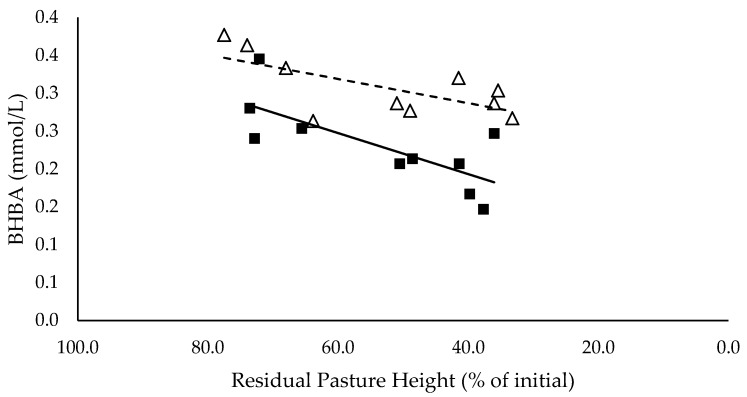
The relationship between the serum β-hydroxybutyrate (BHBA) (mmol/L) and residual pasture height (% of initial) for high (■) and low (Δ) mixed ration levels. Lines were fitted for high (solid) and low (dashed) mixed ration levels.

**Figure 7 animals-10-00904-f007:**
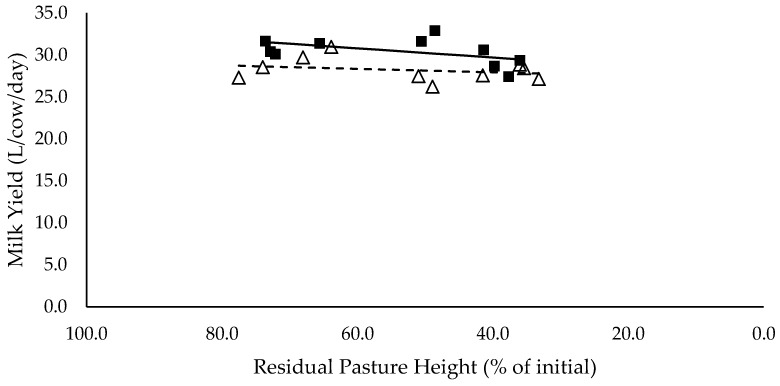
The relationship between milk yield (L/cow/day) and residual pasture height (% of initial) for high (■) and low (Δ) mixed ration levels. Lines were fitted for high (solid) and low (dashed) mixed ration levels.

**Figure 8 animals-10-00904-f008:**
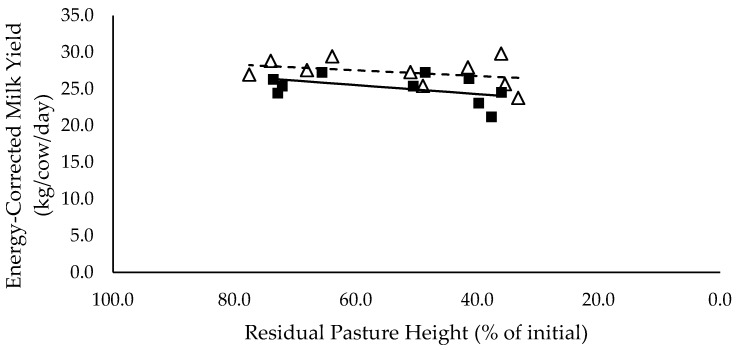
The relationship between energy corrected milk yield (kg/cow/day) and residual pasture height (% of initial) for high (■) and low (Δ) mixed ration levels. Lines were fitted for high (solid) and low (dashed) mixed ration levels.

**Figure 9 animals-10-00904-f009:**
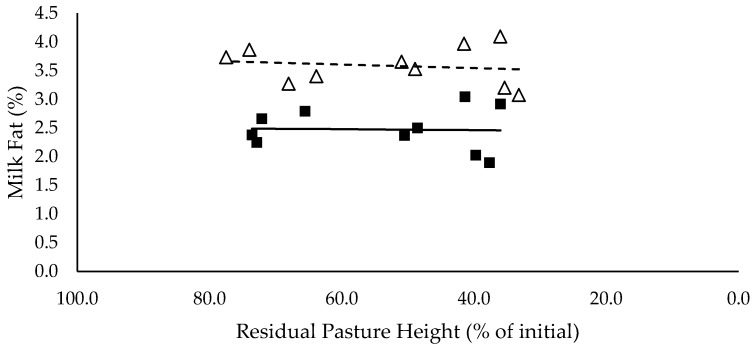
The relationship between milk fat (%) and residual pasture height (% of initial) for high (■) and low (Δ) mixed ration levels. Lines were fitted for high (solid) and low (dashed) mixed ration levels.

**Figure 10 animals-10-00904-f010:**
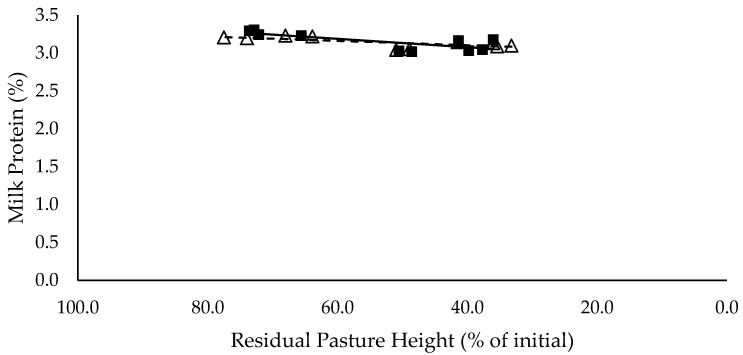
The relationship between milk protein (%) and residual pasture height (% of initial) for high (■) and low (Δ) mixed ration levels. Lines were fitted for high (solid) and low (dashed) mixed ration levels.

**Table 1 animals-10-00904-t001:** Outline of Pasture × Mixed Ration level treatments for thirty multiparous cows.

Mixed Ration Level	Target Residual Height (cm)	Expected Area Grazed (%)	Pasture Allocation Group	Target Pasture Allocation (m^2^/cow/day)	Replicates	Cows per Replicate
High	5	100	1	70	1	3
High	12	100	2	95	1	3
High	19	100	3	120	1	3
High	25	95	4	210	1	3
High	25	80	5	290	1	3
Low	5	100	1	140	1	3
Low	12	100	2	190	1	3
Low	19	100	3	240	1	3
Low	25	95	4	420	1	3
Low	25	80	5	580	1	3

**Table 2 animals-10-00904-t002:** Diet composition for high and low mixed ration levels with targeted pasture intakes.

Ingredient (kg DM/cow/day)	High	Low
Corn silage	5.4	2.8
Wheat grain	5.4	2.8
Canola meal	1.8	0
Lucerne hay	0.9	0.9
Mineral mix	0.5	0.5
Total Mixed Ration	14.0	7.0
Targeted Pasture	7.0	14.0
Total DM Intake	21.0	21.0

**Table 3 animals-10-00904-t003:** Pasture height and defoliation intensity for each treatment group, arranged in increasing pasture allocations (kg dry matter (DM)/cow/day) for each mixed ration level.

Treatment Group	Mixed Ration Level	Pasture Allocation Group	Pasture Allocation (kg DM/cow/day)	Residual Pasture Height (% of Initial)	Area Grazed (%)	Pre-Grazing Pasture Height (cm)	Grazed Residual Pasture Height (cm)
4	High	1	5.0	38.7	99.0	40.5	15.3
9	High	2	8.3	38.7	97.9	39.7	14.7
6	High	3	10.3	49.6	97.6	39.9	19.3
1	High	4	18.6	68.8	75.5	40.0	23.5
7	High	5	25.2	73.2	84.6	39.7	27.3
8	Low	1	12.7	34.3	98.9	42.6	13.8
2	Low	2	15.9	38.8	97.5	39.4	14.7
5	Low	3	19.8	50.0	99.2	40.2	19.9
10	Low	4	35.3	66.0	95.3	40.4	26.0
3	Low	5	49.2	75.8	78.8	40.3	27.9
				R^2^	0.70	0.01	0.82
				Mixed ration effect ^+^			
				Slope (low)	−0.384 **	−0.0057	0.373 ***
				Slope (high)	−0.559 ***	−0.0087	0.326 ***

^+^ Either main effect or interaction with residual pasture height. Significance indicated by ** *p* < 0.01 and *** *p* < 0.001.

**Table 4 animals-10-00904-t004:** The nutrient (% dry matter (DM)) and metabolisable energy (MJ/kg DM) content of the consumed lucerne pasture for each treatment group.

Mixed Ration Level	Pasture Allocation Group	Residual Pasture Height (% of Initial)	Crude Protein	Metabolisable Energy	Neutral Detergent Fibre	Acid Detergent Fibre
High	1	38.7	26.7	8.77	31.9	27.9
High	2	38.7	28.0	8.83	31.8	28.1
High	3	49.5	31.0	9.54	29.0	25.9
High	4	68.8	36.5	10.82	24.0	22.0
High	5	73.2	37.5	11.06	23.1	21.2
Low	1	34.3	26.4	8.44	33.3	29.3
Low	2	38.8	28.0	8.84	31.8	28.0
Low	3	50.0	31.2	9.56	28.9	25.9
Low	4	66.0	35.3	10.58	24.9	22.7
Low	5	75.8	38.2	11.22	22.4	20.7
		R^2^	0.99	0.99	0.98	0.97
		Mixed ration effect ^+^				
		Slope (low)	0.280 ***	0.0657 ***	−0.257 ***	−0.202 ***
		Slope (high)	0.295 ***	0.0665 ***	−0.258 ***	−0.201 ***

^+^ Either main effect or interaction with residual pasture height. Significance indicated by *** *p* < 0.001.

**Table 5 animals-10-00904-t005:** The nutrient (kg dry matter (DM)/cow/day) and metabolisable energy (MJ/cow/day) intake of the consumed pasture for each treatment group.

Mixed Ration Level	Pasture Allocation Group	Residual Pasture Height (% of Initial)	Offered	Intake	Crude Protein	Metabolisable Energy	Neutral Detergent Fibre	Acid Detergent Fibre	Lignin	Starch	Sugar
High	1	38.7	5.03	3.55	1.21	38.5	1.46	1.28	0.38	0.03	0.26
High	2	38.7	8.33	5.83	1.58	50.2	1.90	1.68	0.49	0.03	0.34
High	3	49.5	10.29	6.00	1.79	55.6	1.81	1.61	0.51	0.04	0.35
High	4	68.8	18.62	6.62	2.15	65.6	1.88	1.69	0.57	0.04	0.39
High	5	73.2	25.24	7.91	2.78	82.5	2.10	1.91	0.68	0.05	0.47
Low	1	34.3	12.66	9.63	2.39	74.9	3.48	3.07	0.81	0.06	0.56
Low	2	38.8	15.87	11.24	3.04	96.8	3.70	3.26	0.95	0.07	0.66
Low	3	50.0	19.80	11.36	3.44	106.1	3.39	3.02	0.96	0.07	0.66
Low	4	66.0	35.32	13.69	4.86	146.0	3.72	3.37	1.20	0.09	0.83
Low	5	75.8	49.21	13.79	4.84	142.8	3.71	3.37	1.18	0.08	0.82
		R^2^	0.92	0.93	0.95	0.94	0.94	0.94	0.95	0.95	0.95
		Mixed ration effect ^+^	***	***	***	***	***	***	***	***	***
		Slope (low)	0.861 ***	0.0963 ***	0.0606 ***	1.654 ***	0.00452	0.00679	0.00881 ***	0.000625 ***	0.00610 ***
		Slope (high)	0.482 ***	0.0769 **	0.033 4***	0.923 ***	0.00915	0.00944	0.00575 **	0.000408 **	0.00398 **

^+^ Either main effect or interaction with residual pasture height. Significance indicated by ** *p* < 0.01 and *** *p* < 0.001.

**Table 6 animals-10-00904-t006:** The mixed ration offered and consumed, and nutrient (kg dry matter (DM)/cow/day) and metabolisable energy (MJ/cow/day) intake of the consumed mixed ration for each treatment group.

Mixed Ration Level	Pasture Allocation Group	Residual Pasture Height (% of Initial)	Offered	Intake	Crude Protein	Metabolisable Energy	Neutral Detergent Fibre	Acid Detergent Fibre	Lignin	Starch	Sugars
High	1	38.7	15.64	15.40	2.47	160.6	3.75	2.34	0.40	5.66	0.37
High	2	38.7	15.64	15.01	2.41	156.6	3.66	2.29	0.39	5.52	0.36
High	3	49.5	15.64	15.05	2.41	156.9	3.67	2.29	0.39	5.54	0.36
High	4	68.8	15.64	15.29	2.45	159.5	3.73	2.33	0.40	5.63	0.37
High	5	73.2	15.64	14.92	2.39	155.6	3.64	2.27	0.39	5.49	0.36
Low	1	34.3	7.57	7.43	1.04	75.3	1.96	1.28	0.22	2.89	0.08
Low	2	38.8	7.57	7.39	1.04	75.0	1.95	1.27	0.22	2.88	0.08
Low	3	50.0	7.57	7.44	1.04	75.4	1.96	1.28	0.22	2.89	0.08
Low	4	66.0	7.57	7.45	1.04	75.6	1.97	1.28	0.22	2.90	0.08
Low	5	75.8	7.57	7.40	1.04	75.1	1.95	1.27	0.22	2.88	0.08
		R^2^	0.99	0.99	0.99	0.99	0.99	0.99	0.92	0.99	0.93
		Mixed ration effect ^+^	***	***	***	***	***	***	***	***	***
		Slope (low)	−0.00016	0.00023	−0.00012	−0.0137	0.00016	−0.00005	0.000045	−0.0001	0.000055
		Slope (high)	0.00002	−0.00300	−0.00049	−0.0307	−0.00074	−0.00044	−0.00005	−0.0012	−0.00007

^+^ Either main effect or interaction with residual pasture height. Significance indicated by *** *p* < 0.001.

**Table 7 animals-10-00904-t007:** The total dry matter (DM) offered and consumed, and nutrient (kg DM/cow/day) and metabolisable energy (MJ/cow/day) intake of the total diet for each treatment group.

Mixed Ration Level	Pasture Allocation Group	Residual Pasture Height (% of Initial)	Offered	Intake	Crude Protein	Metabolisable Energy	Neutral Detergent Fibre	Acid Detergent Fibre	Lignin	Starch	Sugars
High	1	38.7	20.67	18.95	3.68	199.1	5.21	3.63	0.78	5.69	0.63
High	2	38.7	23.97	20.84	3.99	206.7	5.57	3.96	0.88	5.56	0.70
High	3	49.5	25.93	21.05	4.21	212.5	5.48	3.90	0.90	5.57	0.71
High	4	68.8	34.26	21.91	4.60	225.0	5.61	4.01	0.96	5.67	0.76
High	5	73.2	40.88	22.83	5.18	238.1	5.74	4.18	1.07	5.54	0.83
Low	1	34.3	20.23	17.05	3.43	150.2	5.44	4.34	1.04	2.95	0.64
Low	2	38.8	23.44	18.63	4.08	171.7	5.65	4.53	1.18	2.94	0.74
Low	3	50.0	27.37	18.79	4.48	181.5	5.35	4.29	1.18	2.96	0.74
Low	4	66.0	42.89	21.14	5.91	221.5	5.69	4.65	1.43	2.98	0.91
Low	5	75.8	56.78	21.19	5.88	217.9	5.66	4.64	1.41	2.96	0.90
		R^2^	0.90	0.78	0.90	0.87	0.17	0.72	0.91	0.99	0.65
		Mixed ration effect ^+^	**	**	**	***		***	***	***	
		Slope (low)	0.861 ***	0.0965 ***	0.0605 ***	1.640 ***	0.00468	0.00673	0.00886 ***	0.00050	0.00616 ***
		Slope (high)	0.482 ***	0.0739 **	0.0330 ***	0.892 **	0.00841	0.00900	0.00570 **	−0.00077	0.00391 *

^+^ Either main effect or interaction with residual pasture height. Significance indicated by * *p* < 0.05, ** *p* < 0.01 and *** *p* < 0.001.

**Table 8 animals-10-00904-t008:** Proportion of pasture, mixed ration and nutrient (% dry matter (DM)) and metabolisable energy (MJ/kg DM) content of the total diet for each treatment group.

Mixed Ration Level	Pasture Allocation Group	Residual Pasture Height	Pasture	Mixed Ration	Crude Protein	Metabolisable Energy	Neutral Detergent Fibre	Acid Detergent Fibre	Lignin	Starch	Sugars
High	1	38.7	0.19	0.81	19.52	10.53	27.60	19.21	4.12	30.09	3.34
High	2	38.7	0.28	0.72	19.12	9.92	26.69	19.00	4.23	26.72	3.37
High	3	49.5	0.28	0.72	19.96	10.10	26.03	18.52	4.26	26.55	3.38
High	4	68.8	0.30	0.70	20.98	10.26	25.60	18.29	4.39	25.91	3.45
High	5	73.2	0.34	0.66	22.65	10.43	25.13	18.29	4.65	24.35	3.63
Low	1	34.3	0.56	0.44	20.27	8.89	31.90	25.45	6.11	17.41	3.79
Low	2	38.8	0.60	0.40	21.88	9.24	30.29	24.22	6.29	15.95	3.94
Low	3	50.0	0.60	0.40	23.84	9.68	28.44	22.81	6.29	15.86	3.94
Low	4	66.0	0.64	0.36	28.12	10.56	27.09	22.14	6.79	14.24	4.33
Low	5	75.8	0.64	0.36	27.77	10.35	26.62	21.79	6.61	14.20	4.23
		R^2^	0.97	0.97	0.89	0.49	0.85	0.94	0.98	0.96	0.60
		Mixed ration effect ^+^	***	***	***		***	***	***	***	***
		Slope (low)	0.000018 *	−0.000018 *	0.190 ***	0.0366 **	−0.120 ***	−0.0819 ***	0.0135 **	−0.0723 *	0.0115
		Slope (high)	0.00003 **	−0.00003 **	0.081 **	0.0055	−0.053 *	−0.0240	0.0110 *	−0.0965 **	0.0062

^+^ Either main effect or interaction with residual pasture height. Significance indicated by * *p* < 0.05, ** *p* < 0.01 and *** *p* < 0.001.

**Table 9 animals-10-00904-t009:** The summary statistics of the linear relationships between all response variables in Figure 1, Figure 2, Figure 3, Figure 4, Figure 5, Figure 6, Figure 7, Figure 8, Figure 9 and Figure 10 and the explanatory variable, residual pasture height (% of initial).

Y Variate	R^2^	High Intercept	High Slope	Low Intercept	Low Slope
Pasture intake (kg dry matter (DM)/cow/day)	0.93	1.84	0.077 **	6.84	0.0963 ***
Total Intake (kg DM/cow/day)	0.78	17.1	0.074 **	14.3	0.0965 ***
Pasture Intake Rate (kg DM/cow/h)	0.84	0.274	0.014 **	0.792	0.0167 ***
Pasture Mincing Energy (kJ/kg DM)	0.97	63.1	−0.4 ***	64.5	−0.440 ***
Non-esterified fatty acid (NEFA) (mEq/L)	0.48	0.829	−0.009 *	0.892	−0.00847 *
β-hydroxybutyrate (BHBA) (mmol/L)	0.72	0.085	0.003 **	0.223	0.00160
Milk Yield (L/cow/day)	0.69	27.6	0.0546	27.1	0.0211
Energy Corrected Milk Yield (kg/cow/day)	0.47	21.9	0.0610	25.2	0.0397
Milk Fat (%)	0.51	2.43	0.0008	3.41	0.00324
Milk Protein (%)	0.24	2.9	0.0055 *	3.0	0.00293

Significance of slopes indicated by * *p* < 0.05, ** *p* < 0.01 and *** *p* < 0.001.

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
