# Peer review of "Maximising Lucerne (Medicago sativa) Pasture Intake of Dairy Cows: 2—The Effect of Post-Grazing Pasture Height and Mixed Ration Level"

_animals, 2020, doi:10.3390/ani10050904_

Round 1

Reviewer 1 Report

The manuscript has been markedly improved and most of my comments have been satisfactorily addressed. However, I still see one major comment/concern regarding the PUP and need to be clarified/addressed. I could see that the focus of the manuscript, but not the discussion, has been redirected away from the PUP strategy; however, the discussion section is still heavily based on the PUP strategy. The PUP areas were estimated visually (as described in the method) which is a very subjective approach and could be associated with bias/error. I understand that that the irregular shapes of trampled and faecal contaminated pasture areas are hard to measure, but how the farmer can know that the 5% of UPU (as suggested in this manuscript) has been achieved.

Additional comments

Line 170 how ether extract was analysed

Line 539-541 move this to the end of milk paragraph (Line 560)

Few typo were spotted, for example

Line 413, 489, 507

Author Response

The manuscript has been markedly improved and most of my comments have been satisfactorily addressed. However, I still see one major comment/concern regarding the PUP and need to be clarified/addressed. I could see that the focus of the manuscript, but not the discussion, has been redirected away from the PUP strategy; however, the discussion section is still heavily based on the PUP strategy. The PUP areas were estimated visually (as described in the method) which is a very subjective approach and could be associated with bias/error. I understand that that the irregular shapes of trampled and faecal contaminated pasture areas are hard to measure, but how the farmer can know that the 5% of UPU (as suggested in this manuscript) has been achieved.

  1. Measuring the exact point along transects for the pre and post grazing measurement provides a very accurate method for determining whether an area was grazed. The method to measure ‘ungrazable areas’, either trampled or spoiled by faecal matter has been expanded within the text (Lines 215 to 228):

    “Defoliation intensity was defined by calculating the vertical and horizontal utilisation of the pasture. Height measurements were taken at fixed assessment points along transects within each grazing strip to allow more accurate measurement of the vertical difference in pasture height pre and post grazing[1,2,6]. Transects were taken every 2 m across the grazing strip, and measurements of pasture height, and condition were recorded every 1 m along each transect, with approximately 100 measurement points per strip per day. The condition of the pasture was assessed at each point visually, a decline in pasture height indicated ‘grazed’ and no change ‘ungrazed’. All assessment points were inspected for feces, and were recorded as 'contaminated' if present. Areas were identified and recorded as 'trampled' when pasture had been flattened onto the soil surface by cows walking or lying down. Trampled and contaminated points were later used to determine the ‘ungrazable’ area within each strip as cows cannot bite into trampled pasture and avoid grazing areas contaminated with feces [2]. The ungrazed area was subtracted from the total area allocated to determine the maximum ‘grazable’ area within each strip. Using the vertical distribution of DM and chemical composition of the pasture outlined in section 2.2, pasture intake and quality was calculated as described by Ison et al. [6]. The combined horizontal and vertical utilisation of the grazed pasture was used to calculate the average intake for each treatment group every day.”
  2. The ‘5% ungrazed pasture’ value was included as a guideline figure. However, we understand that this message is not clear for the reviewer and therefore is unlikely clear for the reader. They key message around the “PUP grazing strategy” is to ensure cows are only grazing the top-leafy stratum of the pasture sward to maximise intake. This is only achieved when some pasture remains un-grazed around the faecal patches. When all the uncontaminated pasture has been consumed, the cows instead of fully grazing the un-grazed pasture around the faecal patches they prefer to re-grazed the bottom stemmy stratum of the uncontaminated pasture and decrease intake. Therefore, faecal patches are good visual indicators of the breaking point of pasture intake which occurs when the cows transition from grazing the top leafy stratum to the bottom stemmy stratum. Therefore we have rectified the manuscript to focus on using faecal patches within the pasture strip as an easily identified tool to optimise allocation and ensure cows are grazing the top-leafy stratum and maximising intake.

This key message has been clarified throughout the manuscript in the abstract, introduction, discussion and conclusion:

Abstract: "In the high allocation treatments, an area of pasture remained un-grazed and cows were only grazing the top-leafy stratum where pasture intake rate and intake were highest. Therefore, to maximise intake in sub-tropical PMR systems, lucerne pasture should be allocated so that cows are always grazing the top-leafy stratum. This can be achieved by ensuring the pasture around faecal patches remains ungrazed." (Lines 35 to 39).

Introduction: "When PUP approaches 0%, intake declines as cows shift from grazing the top leafy stratum to subsequent lower strata where the pasture mass removed declines [6]. Achieving horizontal utilisations where the pasture around faecal patches remains ungrazed (“PUP strategy”), ensures cows are only grazing the top leafy stratum and intake and diet quality are maximized. This can be achieved by allocating pasture based on the top leafy stratum on offer, excluding the proportion of the paddock contaminated with faecal patches. Therefore, the higher the amount of faecal patches, the greater the proportion of the total paddock area that should be left un-grazed to maximize pasture intake" (Lines 70 to 75).

Discussion: "Since the PUP strategy has been shown to drive pasture intake in grazing systems [1,2,6], the results are discussed below in the context of the PUP grazing strategy. PUP has been defined as a grazing strategy to maximise pasture intake in PMR dairy systems [6]. This is achieved by allocating pasture areas large enough to ensure cows selectively consume the top leafy stratum. Cows will remove the top leafy stratum across the whole area of a paddock before regrazing an area, and intake declines. This transition can be observed when cows begin to graze around faecal patches." (Lines 485 to 490).

Conclusion" The ideal post-grazing target for lucerne pastures when grazed at an ideal pre-grazing height (>39 cm above ground level, identified in the companion paper [8]), should ensure the pasture around faecal patches remains ungrazed. Therefore, the higher the amount of faecal patches, the greater the proportion of the total paddock area that should be left un-grazed to maximize pasture intake " (Lines 579 to 582).

Additional comments

Line 170 how ether extract was analysed

  1. Method is referenced in text (line 174).

Line 539-541 move this to the end of milk paragraph (Line 560)

  1. Corrected

Few typo were spotted, for example

Line 413, 489, 507

  1. Errors have been corrected throughout text.

Reviewer 2 Report

Grazing pasture management is often complex by lots of targets of soil, pasture or animals to be identified. This article studies pasture allocation affect pasture utilization, grazing animal intake, milk production and so on. In sub-tropical Australian partial mixed ration (PMR) systems,  offering lucerne with an ideal pre-grazing pasture height and a proportion remains ungrazed, will be benefit and easy for management. The experiement design is reasonble. There are only few comments on the articles.

In 2.1 Experimental design of the M&M, More details about the lucerne pasture should be given, including pasture establishment date, how to manage it? what levels and what kind of fertilizers applied? During grazing, how to prevent the bloating?

In the first paragraph of Discussionit, the proportion of PUP should be fluctuant or range, not exact 5%. In some places, grazing management may increase the density of grazing for weed removal, sometimes, may change grazing stategies from seting grazing to rotation etc. Therefore, how to adjust the ungrazed proportion depends on the pasture quality or stages of growth. 

Author Response

In 2.1 Experimental design of the M&M, More details about the lucerne pasture should be given, including pasture establishment date, how to manage it? what levels and what kind of fertilizers applied? During grazing, how to prevent the bloating?

  1. Some basic details about the pasture management practices used within this study have been included in the text, however specific grazing management and agronomic practices for lucerne will depend on local weather, soil types and rainfall etc:

“The lucerne pasture sward had been established for two years and managed under typical commercial practices for the sub-tropical region [7]. The pasture was fertilized and irrigated to meet the pasture’s nutrient requirements and maintain soil moisture to maximise growth [7]. Bloat oil (271 g/L alcohols, C12-15 ethoxylated) was added daily to water troughs in each paddock at a rate of 42 ml/cow to reduce the risk of bloating” (lines 118 to 122).

In the first paragraph of Discussionit, the proportion of PUP should be fluctuant or range, not exact 5%. In some places, grazing management may increase the density of grazing for weed removal, sometimes, may change grazing stategies from seting grazing to rotation etc. Therefore, how to adjust the ungrazed proportion depends on the pasture quality or stages of growth.

  1. Yes, PUP is not exactly 5% as it depends on the level of faecal contamination. We have clarified this throughout the manuscript. (addressed in response to comment 1  from reviewer 1). As below.

  2. This issue has been corrected throughout the text (addressed in response to comment 1 from reviewer 1):

The ‘5% ungrazed pasture’ value was included as a guideline figure. However, we understand that this message is not clear for the reviewer and therefore is unlikely clear for the reader. They key message around the “PUP grazing strategy” is to ensure cows are only grazing the top-leafy stratum of the pasture sward to maximise intake. This is only achieved when some pasture remains un-grazed around the faecal patches. When all the uncontaminated pasture has been consumed, the cows instead of fully grazing the un-grazed pasture around the faecal patches they prefer to re-grazed the bottom stemmy stratum of the uncontaminated pasture and decrease intake. Therefore, faecal patches are good visual indicators of the breaking point of pasture intake which occurs when the cows transition from grazing the top leafy stratum to the bottom stemmy stratum. Therefore we have rectified the manuscript to focus on using faecal patches within the pasture strip as an easily identified tool to optimise allocation and ensure cows are grazing the top-leafy stratum and maximising intake.

This key message has been clarified throughout the manuscript in the abstract, introduction, discussion and conclusion:

Abstract: "In the high allocation treatments, an area of pasture remained un-grazed and cows were only grazing the top-leafy stratum where pasture intake rate and intake were highest. Therefore, to maximise intake in sub-tropical PMR systems, lucerne pasture should be allocated so that cows are always grazing the top-leafy stratum. This can be achieved by ensuring the pasture around faecal patches remains ungrazed." (Lines 35 to 39).

Introduction: "When PUP approaches 0%, intake declines as cows shift from grazing the top leafy stratum to subsequent lower strata where the pasture mass removed declines [6]. Achieving horizontal utilisations where the pasture around faecal patches remains ungrazed (“PUP strategy”), ensures cows are only grazing the top leafy stratum and intake and diet quality are maximized. This can be achieved by allocating pasture based on the top leafy stratum on offer, excluding the proportion of the paddock contaminated with faecal patches. Therefore, the higher the amount of faecal patches, the greater the proportion of the total paddock area that should be left un-grazed to maximize pasture intake" (Lines 70 to 75).

Discussion: "Since the PUP strategy has been shown to drive pasture intake in grazing systems [1,2,6], the results are discussed below in the context of the PUP grazing strategy. PUP has been defined as a grazing strategy to maximise pasture intake in PMR dairy systems [6]. This is achieved by allocating pasture areas large enough to ensure cows selectively consume the top leafy stratum. Cows will remove the top leafy stratum across the whole area of a paddock before regrazing an area, and intake declines. This transition can be observed when cows begin to graze around faecal patches." (Lines 485 to 490).

Conclusion" The ideal post-grazing target for lucerne pastures when grazed at an ideal pre-grazing height (>39 cm above ground level, identified in the companion paper [8]), should ensure the pasture around faecal patches remains ungrazed. Therefore, the higher the amount of faecal patches, the greater the proportion of the total paddock area that should be left un-grazed to maximize pasture intake " (Lines 579 to 582).

Round 2

Reviewer 1 Report

Comments have been satisfactorily addressed  

This manuscript is a resubmission of an earlier submission. The following is a list of the peer review reports and author responses from that submission.

Round 1

Reviewer 1 Report

The manuscript investigated the effect of post-grazing pasture height of Lucerne and proportion of the mixed ration/pasture in the diet on pasture and total DMI, and milk production of dairy cows.

The manuscript is generally well presented, but few areas need to be clarified before it is accepted for publication. The link between post-grazing residual and what is called “proportion of ungrazed pasture” is unclear, particularly in the method, discussion and conclusion sections. How was the un-grazed area measured? And why its effect was not included in the stat and shown in the result as a separate factor? The PUP grazing strategy is a different name for the grazing at low-stocking intensity, in which animals are offered large grazing area that allow better grazing selectivity to the most digestible part of the plant. However, this may have a long term effect on pasture quality, due to the accumulation of stocky less digestible material, and should be discussed. It has been concluded that Lucerne should be allocated to ensure the PUP does not approach 0%. I really struggled to understand what this conclusion was based on? were different PUP investigated in the study, and 0% was shown optimal? I suggest the conclusion should be nested around the post-grazing residual height rather than PUP. The extra ME intake was not reflected in extra production in this study. Estimating pasture intake for grazing animal is a challenge and method is associated with an error. This could be acknowledged in the discussion as one possible reason for lack of differences in production (e.g. due to DMI being overestimated). Excessive usage of abbreviations in this manuscript resulted in a very boring reading. I suggested that the number of used abbreviations to be minimised.

Summary

Line 14-16 hard to follow

Abstract

Line 27+28 is this pasture allocation above ground level

Line 28 how grazing intensity was measured?

Line 29-30 very vague statement

Line 34-37 is this shown in the results? Which measurement approaching 0%? ambiguous statement     

Intro

Line 45 is PMR= Pasture and mixed ration system? It has been used later in the manuscript to reflect the mixed ration only (e.g. Line 125)

Line 55 “ Cows only grazed the TLS when an area of the paddock remained un grazed” this is an unclear statement

Line 56-68 add a reference

Line 58-59 is this a finding from other study or your hypothesis?

Line 59 which results? From other studies? Add references

Line 60 definition and then abbreviation (e.g. PUP), not the other way around

Line 64-65 how this could be achieved?

Line 75 change to …. in the study of Ison or identified by Ison

Line 77 I still can’t follow what is the PUP and where in your study PUP was compared to “higher pasture utilisation”

Line 79 what is RPH?

Line 76-80 unclear objectives and hypothesis. Your objectives were to investigate the effect of Post-grazing pasture height and mixed ration level on nutrients and DM intake, and milk production and composition of dairy cows

Your hypothesis- pasture and total DMI decline as post-grazing residual decline; milk production declines as post-grazing residual decline; and mixed ration proportion in the diet has no effect on DMI and milk production  

Method

Line 96 I don’t think this is level, but the proportion in the diet. The total allocated diet (DM/cow/day) is 21 kg, comprised of either 33% of mixed ration and 67% pasture or 67% mixed ration and 33% pasture. The pasture was offered at different levels above ground level to achieve the 5 desired post-grazing pasture heights.

Line 104 and Table 1 what do you mean by “Target Area Grazed”. What is the difference between pasture allocation group 4 and 5? this is another factor that has been added to the experiment design?

Line 147 how TDN was measured?

Line 149-150 what do you mean by the “area of utilisation”? is it “Target Area Grazed” reported in table 1? Line 151-152 brief description needs to be added to clarify this method. Was this achieved by offering a bigger pasture area? How was average post-grazing pasture height maintained for pasture allocation group 4 vs 5?

Line 157 was the 25 cm post-grazing height set for optimal PUP? Reference? And explain.

Line 177 why fixed? Why not at random?

Line 181-182 were they subtracted as area from area? How were irregular areas measured? How were the un-grazed areas determined?

Line 239 is the RPH different from “Grazed Residual” in Table 3? Why they are named differently?

Results

Line 240-243 this is a very confusing description! Is RPH% of initial = (Grazed residual (RPH)/pre-grazing pasture height)x 100%? If yes, please simplify your confusing definition. This should have been explained in the method.

Table 3 and other tables Define all abbreviations such as PMR, RPH, and PGPH. The table should stand alone and be self-explanatory. Why interaction and PMR are not split? Split to clearly present whether it is interaction or PMR effect

Line 279 Lignin, starch and sugars are not shown in table 4

Line 295 change to …total DM, CP and ME intake …..increased..

Line 298 table 7 shows that total lignin and sugar intake DID HAVE a significant relationship

Line 300 change to ……,BUT was significantly lower

Line 306 this is not true for lignin and sugar. Check table 8

Line 308-309 add the *** to table 8 to show the PMR effect on starch

Table 4 and other tables check journal style to find out whether you need to define nutrients such as NDF and ADF ..etc

Table 8 change caption to “proportion of Pasture, PMR and ….”

Line 361 change to “P>0.05” if it wasn’t significant

Line 366 add “full stop” before “Average …..”

Line 378 change subheading to Plasma metabolites or blood indices

Line 382 ….than high PMR group

Discussion

Line 427 I’m not sure where this “PUP grazing strategy” is reflected in results. This is about various stocking densities (animal/area) resulting in various post-grazing heights.

Line 436-438 do you mean RPH? Abbreviations are really hard to follow and very confusing

Line 439-440 what is not shown in results?

Line 441 why references listed in a different format

Line 443-446 you pretty much saying that get the animal to graze in a low stocking density to optimise animal selectivity for the most digestible part of the plant. Is this something new? What is the consecutive effect on pasture quality for the second grazing round?

Line 455 reference format

Line 459-467 contradictory statements. The pasture quality limited intake by ADF rather than NDF content of the diet

Line 496-498 but how would you explain the increase in BHBA with the increase of RPH?

Where has that extra ME intake gone? Possibility for errors in DM intake estimation? Quantifying pasture intake for grazing animal always a challenge and methods are associated with errors. This should be acknowledged in your discussion

Conclusion

This is a very long conclusion and should be shortened, focused on key results and deliver concise messages. The whole conclusion should be rewritten

Line 505-507 I’m not sure about this.

Line 515-516 which part of results showing this? This is a confusing and misleading statement

References

There are few references lack of full description and couldn’t be found such as reference 4, 6 and 7 

Reviewer 2 Report

This paper reports the effect of post-grazing pasture height and supplementation of partially mixed rations on intake and performance of grazing dairy cows. The results contribute some new knowledge about grazing management of dairy cattle. However, the following issues need to be addressed:

1.     The necessity of conducting such a study focusing on ‘post-grazing pasture heights’ needs to be justified and the practical implications of the expected results need to be clarified in Introduction. It seems that grazed pastures are seldom allocated for dairy cows to graze further in production practice, thus the above issues need to be addressed.

2.     The paper title suggests that 3 factors (i.e., the proportion of un-grazed pasture, post-grazing pasture heights, and partially mixed rations) are expected to be tested in this study. However, this is not completely conveyed through the experimental design because the authors actually designed a 5 (post-grazing pasture heights) ´ 2 (mixed ration supplementation) factorial experiment (L95-L97), and only these two factors are tested statistically and interpreted in this paper. But when readers look through Table 1, the third factor (‘Target area grazed’) is indeed involved. The inconsistency in the experimental design indicates that this grazing study is not well controlled because any detected statistical differences in intake and animal performance are related to not only pasture heights and mixed ration supplementation, but also to ‘area grazed’ ranging from 75.5 to 99.2% (Table 3).

3.     Although the authors designed a 5 (post-grazing pasture heights) ´ 2 (mixed ration supplementation) factorial experiment, the main effect of the 2 factors as well as their interactions (and corresponding P values) are not completely clarified and presented in either Tables or Figures. Therefore, two-way ANOVA should be carried out for the observations so that the results of this study can be interpreted properly.

4.     Table 2 should present the nutritional composition (determined or calculated) of the TMR and pasture herbages after grazing.

5.     The procedure for estimating pasture intake by dairy cows needs to be depicted because the entry of the key reference (L554) listed at the end of the text is unfortunately incomplete. It appears that the authors adopted an indirect approach to estimate the animal’s pasture intake by analyzing the nutritional composition of the stratified pasture herbages, thus the accuracy of such an indirect method in comparisons with in vivo pasture intake by the animals needs to be justified because pasture intake is a vital parameter for any feeding trials with grazing animals.

6.     Numerous stylistic errors are found in the text. For example, the full name of ‘PMR’ should be defined in title (L3); ‘Ison, et al. [6] …’ (L52) should be ‘Ison et al. [6] …’; it is not correct to define two ‘ME’s (L139 and L143) in a single paper; ‘The quality (Crude Protein, CP % DM; Metabolisable Energy, ME MJ/kg DM; Neutral Detergent Fibre, NDF % DM; Acid Detergent Fibre, ADF % DM; Lignin % DM; Starch % DM and Sugars % DM) …’ (L278-L279) should be read as ‘The quality [crude protein (CP), %DM; metabolisable energy (ME), MJ/kg DM; neutral detergent fibre (NDF), %DM; acid detergent fibre (ADF), %DM; lignin, %DM; starch, %DM; and sugars, %DM] …’.

7.     The Conclusions section needs to be concise. Only significant findings are expected to be summarized here, and any explanatory statements should be moved into Discussion section.

Reviewer 3 Report

Overall comments

I have found this an incredibly difficult paper to read, follow and hold my attention. Papers should be written in such a way that people want to keep reading past the abstract, intro etc.

It is not helped by over use of abbreviation, the title has two, the introduction has 10.

The study seems straightforward, a 2 x 5 factorial but I got completely lost in the description in the methods section.

Table 1 shows 15 cows per PMR which is probably just about enough for production work. However 3 cows per treatment in the factorial design is a very small number of statistical units for production (milk) components to be reported. This is a major flaw, as I can understand, of the study.

Other comments

L20-21 how do you define ‘defoliation intensity’ is it post grazing height or herbage mass

Intro – I have counted 10 abbreviations, some are not used again.

L87 was the pasture regrazed in the second period or was it all freshly grazed over the two 8d measurement periods?

L90 space after bracket

Table 3 Is N=1 for area % grazed, PGPH, grazed residual? Were the pastures replicated or did the 3 cows graze the one paddock/ block?

Table 4 were the 22 pasture samples taken across the PA grazing areas at the beginning of each 8d measurement period or were they taken daily/ weekly etc? It is difficult therefore to know what the data in Table 4,5,6,7,8 represents?

I did not read the discussion as I struggled with the intro, methodology and results section interpretation.